# Folate-Modified Chitosan 5-Flourouraci Nanoparticles-Embedded Calcium Alginate Beads for Colon Targeted Delivery

**DOI:** 10.3390/pharmaceutics14071366

**Published:** 2022-06-28

**Authors:** Shafi Ullah, Asif Nawaz, Arshad Farid, Muhammad Shahid Latif, Muhammad Fareed, Shakira Ghazanfar, Charis M. Galanakis, Abdulhakeem S. Alamri, Majid Alhomrani, Syed Mohammed Basheeruddin Asdaq

**Affiliations:** 1Advanced Drug Delivery Lab, Gomal Centre of Pharmaceutical Sciences, Faculty of Pharmacy, Gomal University, Dera Ismail Khan 29050, Pakistan; shafikustian@gmail.com (S.U.); shahidlatif1710@gmail.com (M.S.L.); fareedwazir05@gmail.com (M.F.); 2Gomal Center of Biochemistry and Biotechnology, Gomal University, Dera Ismail Khan 29050, Pakistan; 3Functional Genomics and Bioinformatics, National Agricultural Research Centre, Islamabad 45500, Pakistan; shakira_akmal@yahoo.com; 4Department of Research and Innovation, Galanakis Laboratories, Skalidi 34, GR-73131 Chania, Greece; cgalanakis@chemlab.gr; 5Department of Biology, College of Science, Taif University, Taif 21944, Saudi Arabia; 6Food Waste Recovery Group, ISEKI Food Association, 1190 Vienna, Austria; 7Department of Clinical Laboratory Sciences, College of Applied Medical Sciences, Taif University, Taif 21944, Saudi Arabia; a.alamri@tu.edu.sa (A.S.A.); m.alhomrani@tu.edu.sa (M.A.); 8Centre of Biomedical Sciences Research (CBSR), Deanship of Scientific Research, Taif University, Taif 21944, Saudi Arabia; 9Department of Pharmacy Practice, College of Pharmacy, AlMaarefa University, Riyadh 13713, Saudi Arabia; sasdag@mcst.edu.sa

**Keywords:** chitosan, 5FU, colon cancer, beads, folate-modified nanoparticles

## Abstract

Gel beads are formed when alginate acid reacts with divalent cations, particularly Ca^2+^. As a result of this feature, it is one of the best materials for making gel beads. Furthermore, it swells only slightly at acidic pH, resulting in stable alginate acid beads, but swells and dissolves rapidly at higher pH values, leading to pH-responsive release. Our current study aimed to embed folate-modified chitosan 5FU nanoparticles (FA-CS-5FU-NPs) into calcium alginate beads for colon-targeted delivery. Calcium alginate beads were developed successfully. Based on the method of drying, two types of beads were obtained: freeze-dried folate-modified chitosan 5FU nanoparticles-embedded beads (FA-CS-5FU-NP-Bf) and oven-dried folate-modified chitosan 5FU nanoparticles-embedded beads (FA-CS-5FU-NP-Bo). The size of (FA-CS-5FU-NP-Bf) was significantly larger than (FA-CS-5FU-NP-Bo). Swelling index (SI), erosion index (EI), and water-uptake index (WUI) of (FA-CS-5FU-NP-Bf) beads were significantly higher than FA-CS-5FU-NP-Bo beads at simulated intestinal pH. An insignificant difference was observed in the release rate of 5FU between (FA-CS-5FU-NP-Bf) and FA-CS-5FU-NP-Bo. The release rate of FA-CS-5FU-NPs was significantly higher than FA-CS-5FU-NP-Bf and FA-CS-5FU-NP-Bo. Pharmacokinetic parameters of 5FU solution, FA-CS-5FU-NPs, and FA-CS-5FU-NP-Bo were analyzed. Solution of pure 5FU showed significantly higher C_max_ and lower AUC, T_1/2_, and Vd than both FA-CS-5FU-NPs and FA-CS-5FU-NPs-Bo, suggesting that FA-CS-5FU-NPs and FA-CS-5FU-NPs-Bo have sustained-release behavior. Biodistribution studies also show that maximum drug amounts were found in the colon from nanoparticles-embedded beads. FA-CS-5FU-NPs-Bo avoid releasing drugs in the stomach and small intestine and make them available in the colon region in higher concentrations to target the colon region specifically.

## 1. Introduction

Although nanoparticles (NPs) prolong the release of drugs, they also improve bioavailability (BA), reduce toxic side effects, and provide a suitable type for all routes of administration [1]. Additionally, ligand-conjugated nanoparticles (NPs) reduce the associated cytotoxic effects of anticancer drugs. Vitamin folic acid (FA) is one of the well-studied ligands used to target the cancer cells. It was reported that uptake of FA into the cells is mediated by the folate receptors (FRs) [2]. In one study, a novel folic-acid-modified 5FU-loaded nanoparticles to target HT-29 cancer cells were designed and showed great efficiency in the cytotoxicity of nanoparticles and cell-specific targeting [3,4]. It was reported that folate-modified liposomes loaded with 5FU exhibited high in vivo antitumor activity as compared with free drug [5]. 5FU-loaded nanoparticles modified with folic acid were reported to significantly enhance the accumulation in tumor cells in vitro; meanwhile, it also resulted in an improved efficacy of folate-conjugated nanoparticles in a mouse model with colon cancer [6]. Despite advantages of NPs, they cannot prevent the drug from being completely released into the gastric medium. In some cases, such as colon cancer treatment, the premature release of medication into the upper portion of GIT and inefficient drug release at the tumor site from the nanocarriers can be barriers that may significantly reduce efficacy. To overcome these barriers, various pH-sensitive polymers have been investigated [7]. As polysaccharide-based materials offer significant advantages in terms of biodegradability, biocompatibility, adhesion, safety, and gelation properties [8], among those polysaccharides, marine derived polysaccharides have various advantages of being widely distributed, abundant in content, simple to manufacture, biodegradable, and biocompatible [9]. Alginate is one of them, showing excellent mucosal adhesion, biodegradability and biocompatibility [10].

Nanoparticles containing the drug for colon cancer (CC) are embedded into sodium alginate beads to avoid the early release of anticancer agent into the stomach and make it available in the colon region. Alginate acid is a natural linear polysaccharide isolated from brown algae [11]. It is composed of β-D-mannuronic acid and α-L-guluronic acid residues linked by β-1,4 glycosidic bonds. Gel beads are formed when alginate acid reacts with divalent cations, particularly Ca^2+^. As a result of this feature, it is one of the best materials for making gel beads. Furthermore, it swells only slightly at acidic pH, resulting in stable alginate acid beads, but swells and dissolves rapidly at higher pH values, leading to pH-responsive release [12]. Previously, both chitosan and alginate acid were employed to fabricate delivery tools that better protect and control the release profile of insulin. Chitosan-coated alginate nanoparticles and microbeads reduce insulin release in simulated gastric buffer but accelerate the release when in simulated intestinal conditions [13].

The current study aimed to embed folic-acid-modified nanoparticles into alginate beads for colon-targeted delivery. The purpose of embedding NPs is: Firstly, the beads protect the drug from premature release into the upper GI portion, especially the gastric medium. Secondly, upon dissolution and ionization in the intestinal medium, the alginate has mucoadhesive property [14]. Most of mucoadhesive systems are prepared out of positively charged polymers such as chitosan and chitosan-coated sodium alginate [15]. These systems provide an intimate contact with the negatively charged (due to sialic acid or carboxyl or sulphate groups in the mucus glycoprotein) mucus membrane due to polyvalent adhesive interaction or electrostatic attraction, H-bond formation, van-der-Waal forces, and others [16]. These phenomena result in further prolongation of nanoparticles’ intestinal residence time. Previously, chitosan-coated nanoparticles were coated with alginate, but folic-acid-conjugated chitosan nanoparticles embedded in alginate beads are being studied for the first time. This study aimed to design folate-modified chitosan 5FU nanoparticles-embedded calcium alginate beads for colon-targeted delivery.

## 2. Materials and Methods

### 2.1. Materials

Chitosan (DD, 83% and mol wt 310,000–375,000) and sodium alginate (Mw = 710,974) were obtained from Sigma-Aldrich (Saint Louis, MO, USA, lot# A263299). 5-flourouracil and folic acid were obtained from Sigma Aldrich (Saint Louis, MO, USA). TPP (85%), potassium dihydrogen phosphate, calcium chloride, carbodiimide (EDC), and sodium hydroxide were obtained from Sigma Chemicals (Saint Louis, MO, USA). Acetic acid, hydrochloric acid, ethanol, and dimethyl sulfoxide (DMSO) were purchased from Merck (Darmstadt, Germany). All the chemicals used were of analytical grade.

### 2.2. Conjugation of Folate–Chitosan (FA-CS)

First, FA and 1-Ethyl-3-(3-dimethyl aminopropyl) carbodiimide (EDC) solution in anhydrous dimethyl sulfoxide (DMSO) (20 mL), with a 1:1 molar ratio, was made and stirred at room temperature until EDC and FA were mixed well. The solution was then slowly added to 0.5% (*w*/*v*) CS in an aqueous solution of 0.1 M of acetic acid having a pH of 4.7, then stirred at 25 °C in the dark area for 16 h to let FA conjugate onto CS molecules. Next, 1 M NaOH was added to adjust the pH of the solution to 9.0. Next, the solution was centrifuged at 2500 rpm to settle the FA-CS conjugate. Next, the sediment was first dialyzed against phosphate buffer having pH 7.4 for 3 days and then against water for 4 days. Finally, the FA-CS conjugate was collected as a sponge by freeze-drying and kept for further study [17]. Chitosan reaction with folic acid is explained in Figure 1.

### 2.3. Characterization of CS-FA Conjugate

#### 2.3.1. Fourier Transform Infrared Spectroscopy

The Fourier transform infrared spectroscopy (FT-IR) was performed using an ATR FTIR spectrometer (L1600300, PerkinElmer, Buckinghamshire, UK). FT-IR spectra of chitosan, folic acid, and conjugate (FA-CS) were obtained. The recording range of the spectrum was 600–4000 cm^−1^ at 32 scans per minute with a resolution of 4 cm^−1^ in absorbance mode. After recording, the spectra were baseline-corrected and normalized using Spectra software to identify the characteristic peaks and differences [18].

#### 2.3.2. Determination of Folic Acid (FA) Content

First, FA-CS conjugates were accurately weighed and then dissolved in 50 mL of 0.2 M sodium bicarbonate buffer solution having a pH of 10 at 25 °C with magnetic stirring. Next, the solution was centrifuged at 3500 rpm for 10 min (Laboratory centrifuge, YJ03-0434000, Shanghai, China). After that, the supernatant was tested to determine folate (FA) using a UV visible spectrophotometer with a wavelength of 365 nm. The folate content was calculated as the percentage of FA in a unit weight of conjugate. At least three replicates were carried out for each experiment, and the results were averaged.

### 2.4. Preparation of Nanoparticles (NPs)

FA-CS-NPs were prepared by the ionic cross-linking method, as mentioned by Salar and Kumar, 2018 [19], with slight modifications. Tripolyphosphate (TPP) was used as a cross-linker. FA-CS conjugate solution (0.2%, *w*/*v*, pH 2.5) was prepared using 1% *v*/*v* acetic acid at room temperature. TPP (0.2%, *w*/*v*) solution in distilled water was prepared. For the synthesis of 5FU-loaded nanoparticles, an aqueous solution of 5FU (500 mg/10 mL) was prepared separately. Solution of 5FU was added dropwise into the FA-CS conjugate solution. TPP solution was added into conjugate solution dropwise in the ratio of 1:3. The solution was allowed to stir for 1 h on a magnetic stirrer at room temperature. NPs suspension was centrifuged at 5000 rpm for 10 min to separate the NPs from the solution and then freeze-dried for 24 h to obtain the final product of NPs. NPs without FA conjugation were also prepared similarly [19].

### 2.5. Characterization of Nanoparticles

#### Size and Zeta Potential

Photon correlation spectroscopy was used to determine the particle size and zeta potential of FA-CS-5FU-NPs and CS-5FU-NPs at 25 °C in a quartz and zeta potential cell with a detectable angle of 90°, respectively (Malvern Panalytical Zetasizer Nano ZS 90, Malvern, UK). In 5 mL of deionized water, 1 mg of NPs was added, and vortex stirring (Velp Scientifica, Usmate Velate, Italy) was used for uniform mixing [20].

### 2.6. Preparation of Nanoparticles Embedded in Calcium Alginate Beads

Beads were formulated by the method described by Mandal et al., 2010 [21]. The aqueous solutions of 10 g of a 2–2.5% *w*/*v* sodium alginate containing 100 mg of FA-CS-5FU-NPs (equivalent to 46 mg 5FU) were prepared. The prepared solution was introduced dropwise from a glass syringe with a needle of 22-gauge size into 100 mL of an aqueous calcium chloride solution, being stirred at 350 rpm. The concentration of calcium chloride in the solution was kept constant, i.e., 3% *w*/*v* for alginate solution. The solution was stirred for one hour after adding the last drop of sodium alginate solution. The calcium alginate beads were collected and washed three times with distilled water before being dried in an oven and freeze dryer to obtain FA-CS-5FU-NP-Bo and FA-CS-5FU-NP-Bf, respectively. The process was carried out for 24 h [21].

### 2.7. Characterization of Beads

#### 2.7.1. ATR-FTIR Analysis of Beads

Individual beads were cut into two pieces with a blade. Next, the Fourier transform infrared spectroscopy (FT-IR) was performed using an ATR-FTIR spectrometer (L1600300, PerkinElmer, Buckinghamshire, UK). The recording range of the spectrum was 600–4000 cm^−1^ at 32 scans per minute with a resolution of 4 cm^−1^ in absorbance mode. After recording, the spectra were baseline-corrected and normalized using Spectra software to identify the characteristic peaks and differences.

#### 2.7.2. Bead Size and Shape

Ten beads from both FA-CS-5FU-NP-Bf and FA-CS-5FU-NP-Bo were randomly selected. A digital vernier caliper was used to measure the size and shape of each type of bead (Mitutoyo, Kawasaki, Japan). The mean bead diameter was determined by selecting ten beads randomly. The length and breadth of each bead were measured, and the mean was calculated as their diameter. The roundness of the beads was calculated using the sphericity factor (SF) and aspect ratio (AR). Equation (1) was used to determine the SF.
SF = (d_max_ − d_min_)/(d_max_ + d_min_)(1)
where d_max_ and d_min_ represented the maximum length and minimum breadth, i.e., diameters of the bead, respectively; the SF value ranged from 0 for a perfectly symmetrical bead around its center to close to a unit value for an irregularly shaped bead. The AR parameter was defined as the quotient of maximum diameter to a minimum diameter of beads as given in Equation (2):AR = d_max_/d_min_(2)

For asymmetrical bead, the value of AR is 1, and it rises as the bead length increases. A random sample of at least 10 beads was used for each batch prepared to characterize SF and AR parameters.

#### 2.7.3. Surface Morphology

The surface morphology of FA-CS-5FU-NP-Bf and FA-CS-5FU-NP-Bo were analyzed using a scanning electron microscope (JSM5910, JEOL, Akishima, Japan) and fluorescence microscope (SWIFT, M3300-D, LA, USA).

#### 2.7.4. Determination of Drug Content, Yield, and % Entrapment Efficiency

Aiming to determine the drug content of the beads, a method adopted by Zhang et al., 2010 was used. Drug content and %EE were determined for FA-CS-5FU-NP-Bf and FA-CS-5FU-NP-Bo (equivalent to 5 mg of 5FU). First, beads formulations were stirred at 200 rpm for 4 h at 25 °C in phosphate buffer pH 6.7. The solution was then centrifuged (Ultracentrifuge Optima LE-80K, Beckman Coulter, Pasadena, CA, USA) at 5000 rpm for 30 min at 4 °C; the supernatant was analyzed for the presence of 5FU by UV technique to determine the free drug. Next, the beads were subjected to grinding in mortar and pestle and stirred in dilute hydrochloric acid (HCl) followed by filtration. UV was then used to analyzed the filtrate to determine %EE using the following equation [22].
Encapsulation efficiency (%) = Mi − Md/Mi × 100%(3)
where Mi is the initial weight of drug dissolved in the alginate solution in the form of NPs, and Md is the weight of drug measured in the gelling media right after the preparation of the drug-loaded beads. Triplicates for each batch of beads were conducted, and the results were averaged.
Yield (%) = weight of dry beads/weight of drug and polymers × 100(4)

#### 2.7.5. Bead Swelling, Erosion, and Water Uptake

The length and width of ten beads from FA-CS-5FU-NP-Bf and FA-CS-5FU-NP-Bo were measured using a digital vernier caliper, and their weights were calculated. To simulate stomach conditions, each type of bead was placed in 20 mL of pH 1.2 buffers and agitated for 2 h at 50 rpm and 37 °C. After 2 h, beads were shifted to a simulated intestinal medium with a pH of 6.5. The beads were tested for swelling, erosion, and water uptake for 6 h in the intestinal medium. After eliminating the moisture content from each wet bead, the size and weight were measured by gently sliding the bead over a dry Petri dish until no indication of wetness remained in its immediate area on the dish surface. The bead was then dried in the oven for 24 h at 40 °C before equilibrating to a constant weight in a desiccator at 25 °C. The swelling (SI), erosion (EI), and water uptake (WUI) of each bead are given in the following equations [14].
SI = (St − Si)/Si × 100%(5)
where Si = initial dry bead diameter, and St = wet bead diameter at time t.
EI = Wi − Wt (d)/Wi × 100%(6)
where Wi = initial dry bead weight, and Wt(d) = dry weight of bead collected at t.
WUI = Wt − Wt (d)/Wt (d) × 100%(7)
where Wt = wet weight of bead at t.

#### 2.7.6. In Vitro Release of Beads

The release study was performed separately for FA-CS-5FU-NP-Bf and FA-CS-5FU-NP-Bo and FA-CS-5FU-NPs. The experiment was conducted for up to 24 h using a dissolution tester (Dissolution, type 1 (basket method) in 900 mL and stirred at 37 ± 0.5 °C (first in a pH 1.2 phosphate buffer solution (prepared by dissolving 6.90 g KH_2_PO_4_ and 2 M HCl in 1000 mL distilled water)) for 2 h and then in pH 6.5 phosphate buffer solution up to 24 h). The stirring speed was set at 100 rpm. Amounts of both types of beads equivalent to 7.35 mg of 5FU were placed in a separate basket, and the apparatus was started. After predetermined time intervals (0, 30, 1, 1.5, 2, 4, 8, 12, 16, 20, and 24 h), a 5 mL sample was withdrawn and replaced with a fresh dissolution medium. After the first 2 h, the sample was quickly filtered through a 100-mesh screen, blotted to remove the surface fluid, and then transferred into the pH 6.5 solution for further testing. After appropriate dilution, the samples were analyzed using a UV spectrophotometer at 265 nm. The cumulative percentage of the drug release was calculated [23].

### 2.8. Pharmacokinetic Analysis

Male Sprague Dawley rats weighing 200 to 250 g were used in this study. The rats were acclimatized for 7 days in individual housing under 12 h light/dark cycles with tap water and standard food ad libitum. The ambient temperature was set at 25 ± 2 °C. The study was conducted according to the Declaration of Helsinki and approved by the Institutional Review Board and Ethics Committee of Gomal University (protocol code No:117/ERB/GU and 26 February 2021). All rats were fasted overnight before the experiments, with free access to water. The rats were divided randomly into three groups with five animals each (*n* = 5). 5FU solution and formulations of FA-CS-5FU-NPs and FA-CS-5FU-NP-Bo were administered by oral gavages at a single dose of 50 mg/kg.

#### Blood Collection and HPLC Analysis

The rats were anesthetized by ketamine–xylazine intraperitoneal injection (15 mg ketamine/200 g rat and 2 mg xylazine/200 g rat) to collect blood samples. Blood samples (0.5 mL) were taken from the retro-orbital vein into heparinized micro tubes at the following times: 0.5, 1, 1.5, 2, 4, 6, 8, 12, 16, and 20 h. Male Sprague Dawley rats weighing 200 to 250 g were used in this study. The rats were acclimatized for 7 days in individual housing under 12 h light/dark cycles with tap water and standard food ad libitum. The ambient temperature was set at 25 ± 2 °C. The study was conducted according to the Declaration of Helsinki and approved by the Institutional Review Board and Ethics Committee of Gomal University (protocol code No:383/GCPS/ERB/GU and 26 February 2021). All rats were fasted overnight before the experiments, with free access to water. The rats were divided randomly into three groups with five animals each (*n* = 5). 5FU solution and formulations of FA-CS-5FU-NPs and FA-CS-5FU-NP-Bo were administered by oral gavages at a single dose of 50 mg/kg, and 24 h after dosing. The blood samples were centrifuged at 3020 rpm for 10 min. The supernatant was collected, transferred to Eppendorf tubes, and stored at −20 °C until analysis by HPLC [24].

The released concentration of 5FU was measured using HPLC (HP1100 Liquid Chromatograph, Agilent, Santa Clara, CA, USA). Analytical column Hypersil C18 (5 μm, ID 4.6 mm × 300 mm) was used. Phosphate buffer (0.01 mol/L) was used as a mobile phase, and an elution rate of 1.0 mL/min was kept at room temperature. Absorbance at 265 nm was monitored, and different pharmacokinetic parameters were determined from the obtained absorbance-time curves. The elimination rate constant (Ke) was estimated by a linear regression analysis of the terminal portion of the log-linear blood concentration-time profile of 5FU. The terminal elimination half-life (t_1/2_) was calculated from Ke using the formula t_1/2_ = 0.693/Ke. The maximum observed plasma concentration (C_max_) and the time taken to reach it (T_max_) were obtained from curve plotting 5FU concentration vs. time. The linear trapezoidal rule calculated the area under each drug concentration-time curve (AUC, ng·mL-1·h) to the last data point. The apparent volume of distribution (VD) was calculated by the equation VD = dose/Cp.

### 2.9. Biodistribution Studies

For biodistribution studies, rats were divided into three groups (*n* = 3 rats/group). Each group was orally administered with 5FU solution as controls, nanoparticles, and its beads formulation as test at a dose of equivalent to 50 mg/kg body weight. At 12 and 24 h after oral drug administration, the rats were sacrificed. Heart, liver, lung, kidney, and colon samples were surgically collected. Tissues samples after weighing were washed with ice-cold 0.9% saline and dried with filter paper. Then, these tissues were homogenized with 0.9% saline in appropriate concentration. The homogenates were then centrifuged at 10,000 rpm for 5 min. The supernatant was taken, processed (Section 2.7.1), and analyzed using HPLC [25].

### 2.10. Data Analysis and Statistics

All in vitro results were expressed as the mean ± standard deviation (SD) of three replicates. The in vivo results were presented as the mean ± SD of five replicates. Pharmacokinetic parameters were estimated using the model-independent method. The obtained data were statistically analyzed using ANOVA (one-way analysis of variation) and Student’s *t*-test (IBM^®^ SPSS^®^ Statistics version 19, Armonk, NY, USA) and Statistical Package Minitab^®^ version 20 (Minitab, LLC, State College, PA, USA).

## 3. Results and Discussion

### 3.1. Conjugation of Folic Acid with Chitosan

Degree of substitution (DS) value of FA to monosaccharide residue of CS was found to be 10.5%. FT-IR study confirmed the conjugation of folic acid with chitosan (Figure 2). In FT-IR of chitosan, a strong band at 3400 cm^−1^ represents NH functional groups (primary amine). The absorption band at around 2977 cm^−1^ can be attributed to CH symmetric stretching. Bands at 1401 cm^−1^ indicate the methyl group (CH_3_). Symmetrical bending in the range of 1260–1800 cm^−1^ belongs to the glycosidic ring; in particular, the band at 1156 cm^−1^ corresponds to the glycosidic linkage. Similarly, FT-IR of pure folic acid showed an IR spectrum at 3100–3500 cm^−1^ attributed to OH carboxylic of glutamic acid moiety and NH group of pterin ring stretching. Absorption at 1760 cm^−1^ represents C=O carboxylic. Similarly, the absorption band at 1432 cm^−1^ represents the phenyl and pterin ring. The band at 3321 cm^−1^ pure folic acid is absent/overlapped in conjugate formulation, indicating the coupling of folate with the chitosan polymer [26]. The amide bond formation between the chitosan and folic acid is evidenced by a shift of the FTIR wave number of folic acid at 1760 to 1680 cm^−1^. The assignment of FTIR peaks bands was correlated with earlier studies [27,28].

### 3.2. Preparation of FA-CS-5FU-NPs

Nanoparticles were prepared successfully using the ionic gelation method. The size and zeta potential of FA-CS-5FU-NPs was found to be 235 ± 12 nm and 26 ± 2, respectively. In addition, the polydispersity index was found to be 0.25. Higher zeta potential and lower PDI value indicate that nanoparticles are stable when dissolved in solution.

### 3.3. Beads Encapsulation of FA-CS-5FU-NPs

Although oral formulations of 5FU have fewer side effects than parenteral formulations, the resistance problem of 5FU requires its high dose, which leads to serious adverse GI reactions [24]. Furthermore, its absorption in the stomach is rapid, leading to a decrease in mean retention time and T_1/2_ [29]. Our previous study depicted that FA-CS-5FU-NPs have a premature release in the stomach [30]. Aiming to reduce or eliminate its stomach absorption and make it available in the colon region, its formulation was embedded successfully in calcium alginate beads (Figure 3).

### 3.4. Characterization of Prepared Beads

#### 3.4.1. FT-IR Study

In FT-IR of chitosan, a strong band at the region of 3321.9 cm^−1^ represents OH and NH functional groups and intermolecular hydrogen bonding. The absorption band at around 2977 cm^−1^ can be attributed to CH symmetric stretching. Those at 2908 cm^−1^ represent OH. Broadband at 1450–1500 cm^−1^ attributed to C=C the single substituted benzene ring. Bands at 1401 cm^−1^ indicate the methyl group (CH_3_). Symmetrical bending in the range of 1260–800 cm^−1^ belongs to the glycosidic ring; in particular, the band at 1156 cm^−1^ corresponds to the glycosidic linkage. Alginate displayed vibrations at 3321–3007 cm^−1^ (O–H stretch). The bands at 1404 cm^−1^ were assigned to axial symmetric deformation of carboxylate anions (COO^−^). The symmetric COO stretch peak at 1404 cm^−1^ also shifted to 1450 cm^−1^ in beads (Curve B). This indicates that Ca^2+^ ions reacted with carboxyl groups of sodium alginate to form calcium alginate beads [31]. In addition, the peak at 1450 cm^−1^ is of low intensity compared to the peak at 1404 cm^−1^ in alginate spectra (Curve A). This is due to the complexion among these carboxylate anions with positively charged sites of chitosan molecules [32]. The band at 1050 cm^−1^, attributed to the stretching of C-O bonds of primary alcohols in alginate (Curve A), shifted to 1060 cm^−1^ in the FTIR spectra of the formulation (Curve B). These differences were mainly due to associations among chain segments of alginates and chitosan molecules (Figure 4) [32].

#### 3.4.2. Size, Shape and Surface Morphology

Two types of FA-CS-5FU-NPs-embedded beads were obtained based on their drying method: freeze-died and FA-CS-5FU-NP-Bo. The sizes, shapes, and surfaces of FA-CS-5FU-NP-Bf and FA-CS-5FU-NP-Bo were compared. The mean diameter of freshly prepared wet beads was 2.5 mm. Table 1 shows the sizes and shapes of FA-CS-5FU-NP-Bf and FA-CS-5FU-NP-Bo. The beads of FA-CS-5FU-NP-Bf depicted almost retained their original diameter from before their drying. Its mean diameter was 2.01 mm ± 0.07, which was near-spherical in shape. Its sphericity factor (SF) was 0.16 ± 0.03, and its aspect ratio (AR) was 1.40 ± 1.01. FA-CS-5FU-NP-Bo beads were significantly shrunken (*t*-test, *p* < 0.05) as compared to (FA-CS-5FU-NP-Bf) beads, and their mean diameter was found to be 1.12 ± 0.02, and their shape was also near-spherical. Its sphericity factor (SF) was 0.17 ± 0.07, and its aspect ratio (AR) was 1.5 ± 1.2. FA-CS-5FU-NP-Bf beads have rough surfaces (Figure 5), and at high magnification, the surface of FA-CS-5FU-NP-Bf beads is porous, as freeze-drying enhances the porosity [33]. The surface of FA-CS-5FU-NP-Bo beads is nonporous, smoother, and uniform compared to FA-CS-5FU-NP-Bf beads but has cracks on its surface due to the polymer’s collapsed network structure during drying (Figure 5). Oven drying causes the shrinkage of solid material by capillary pressure because of the high water-surface tension. The shrinking process leads to densification [34]. Due to these factors, the FA-CS-5FU-NP-Bo beads are denser, harder, and less porous than (FA-CS-5FU-NP-Bf) beads, as previously reported for chitosan–lysine beads [35]. Comparatively, in the freeze-drying process, the direct evaporation of frozen samples without transferring through the liquid state reduces the shrinkage of the beads significantly (*t*-test, *p* < 0.05) to develop beads with a nearly unchanged diameter (3 mm instead of 2.5 mm) (Figure 5).

Figure 4 shows the surface morphology of beads. The surface morphology of FA-CS-5FU-NP-Bf beads revealed a more porous structure than FA-CS-5FU-NP-Bo, while the FA-CS-5FU-NP-Bo possesses cracks on their surfaces. In addition, FA-CS-5FU-NP-Bf was nearly spherical, while FA-CS-5FU-NP-Bo was somewhat disc-shaped.

#### 3.4.3. Entrapment Efficiency, Drug Content, and % Yield

The percent EE of FA-CS-5FU-NP-BF and FA-CS-5FU-NP-Bo was found to be 96.2 ± 5 and 92 ± 2, respectively. This difference was insignificant (*t*-test, *p* > 0.05). Drug content for FA-CS-5FU-NP-Bf and FA-CS-5FU-NP-Bo was found to be 43 ± 2 and 38.5 ± 2 (Table 2), which were also insignificantly different (*t*-test, *p* > 0.05) from each other. The yield of FA-CS-5FU-NPs embedded beads was found to be 86.5%.

#### 3.4.4. Bead Swelling, Erosion, and Water Uptake

FA-CS-5FU-NP-Bf and FA-CS-5FU-NP-Bo were subjected separately to investigate their swelling, erosion, and water uptake. The swelling index (SI) of FA-CS-5FU-NP-Bf and FA-CS-5FU-NP-Bo was 1 and 0.5% at pH 1.2, respectively. This difference was insignificant (*t*-test, *p* > 0.05). At pH 6.5, SI of FA-CS-5FU-NP-Bf was (5.5 ± 0.3% to 16 ± 1.5%) significantly higher (*t*-test, *p* < 0.05) than FA-CS-5FU-NP-Bo (2.1 ± 0.5 to 11 ± 2.1). At low-pH values (<4), the carboxylate groups of alginate were protonated [36]. The decrease in the electrostatic repulsion among these groups could give stability to beads. In the acidic fluid, the beads maintained their spherical shape. They floated on the liquid surface, and this was due to the intra-molecular and inter-molecular hydrogen bond between -COOH and -OH groups [31]. The high swelling ratio in-simulated intestinal fluid is because, with the increase of pH, the repulsive force produced by the deprotonated carboxyl group (-COO^−^) of alginate resulted in a higher swelling ratio [31]. The FT-IR study confirmed the protonation and deprotonation of a carboxyl group at simulated gastric and intestinal pH (Figure 6). The FTIR wave numbers of beads at 3339 ± 8.1 cm^−1^ and 1597 cm^−1^ are attributed to the O-H/N-H functional group and C=O asymmetric stretch of COOH and COO^-^ groups, respectively. In the simulated gastric and intestinal fluid medium, peaks were shifted to 1600 cm^−1^ and 1597 cm^−1^, respectively. These changes may be due to protonation and deprotonation in simulated gastric and intestinal media.

At acidic pH, percent erosion of FA-CS-5FU-NP-Bf and FA-CS-5FU-NP-Bo beads was found to be 1.3 ± 0.5 and 1.2 ± 0.7%, respectively, which are insignificantly different (*t*-test, *p* > 0.05) from each other. At simulated intestinal pH, the erosion of FA-CS-5FU-NP-Bf beads was significantly higher than FA-CS-5FU-NP-Bo beads, which were found to be in the range of 7.5 ± 2 to 17.2 ± 1.5 and 2.5 ± 1.5 to 14 ± 1.7, respectively. The FA-CS-5FU-NP-Bo was generally found harder, smaller, and less fragile. This property leads to decrease erosion. The high porosity of beads favored polymers to dissolve in the media. Additionally, the area of contact of polymers with dissolution media increases, leading to increase dissolution of polymers in the media [37]. These factors favor a higher erosion rate of FA-CS-5FU-NP-Bf.

The water-uptake index (WUI) of FA-CS-5FU-NP-Bf and FA-CS-5FU-NP-Bo beads was 4.4 ± 1.2 and 2.2 ± 0.5%, respectively, at acidic pH. At simulated intestinal pH, the percent water uptake of FA-CS-5FU-NP-Bf and FA-CS-5FU-NP-Bo were in the range of 30.3 ± 2 to 70.4 ± 2% and 16.3 ± 2 to 51.2 ± 3%, respectively. WUI of FA-CS-5FU-NP-Bf beads was significantly higher (*t*-test, *p* < 0.05).

#### 3.4.5. In Vitro Drug Release Study

The FA-CS-5FU-NP-Bf and FA-CS-5FU-NP-Bo loaded with 2% (*w*/*w*) of 5FU were subjected to simulated gastric pH 1.2 and intestinal pH 6.5 to investigate the release behavior of 5FU (Figure 7). It was found that there was no release from both FA-CS-5FU-NP-Bf and FA-CS-5FU-NP-Bo at acidic pH. However, at simulated intestinal pH, the rate of the drug released from both types of beads was increased. This is consistent with the swelling character of beads, as our swelling study shows that the percent swelling degree of beads increases with an increase in the pH of the medium. Although there was slight swelling of beads at acidic pH, that swelling was too small to cause drug release from FA-CS-5FU-NP-Bf and FA-CS-5FU-NP-Bo. At simulated intestinal pH, percent release rates of 5FU from both FA-CS-5FU-NP-Bf and FA-CS-5FU-NP-Bo beads ranged between 39.3 ± 2 and 96.5 ± 1 and 20.2 ± 2 and 95.9 ± 3, respectively. FA-CS-5FU-NP-Bf and FA-CS-5FU-NP-Bo beads were disintegrated completely after 8 and 6 h, respectively. The release rate from FA-CS-5FU-NP-Bf from 3 to 8 h was significantly higher (*t*-test, *p* < 0.05) than FA-CS-5FU-NP-Bo. This might be due to the porous nature of FA-CS-5FU-NP-Bf. Due to its porous nature, water-uptake behavior is high, as shown in our study (Figure 8), leading to a high swelling degree. The probable reason for a higher swelling degree at high pH might be due to the electrostatic repulsion between negatively charged carboxyl groups of folic acid and sodium alginate [38]. High rate of swelling causes the beads to release a large amount of drug. The minimal swelling ratio in the case of FA-CS-5FU-NP-Bo was beneficial for restraining the rate of drug release. These observations show that beads can be used to deliver 5FU specifically to the colon without the premature release of the drug in the stomach. The FA-CS-5FU-NP-Bo may be preferably suitable for colon-targeted drug delivery.

### 3.5. In Vivo Pharmacokinetic (PK) Study

A pharmacokinetic (PK) study (Figure 9a) shows the plasma concentrations (µg) versus time profile of 5-fluorouracil after oral administration of FA-CS-5FU-NP-Bo and pure 5FU solution to Sprague Dawley rats at a dose of 50 mg/Kg. The concentration of 5FU from the 5FU solution was detected very soon, and 14.03 ± 1.52 µg C_max_ was obtained in T_max_ of 1 h followed by a decline in concentration up to 4 h (Figure 4) and (Table 3). FA-CS-5FU-NP-Bo and FA-CS-5FU-NPs showed significant differences from the 5FU solution in the pharmacokinetic parameters, such as t_1/2_, AUC_0–∞_ C_max_, and T_max_. Significantly (ANOVA, *p* < 0.05) higher C_max_ and lower Vd of 5FU solution than FA-CS-5FU-NP-Bo and FA-CS-5FU-NPs suggest that a high quantity of 5FU from solution was absorbed in the upper portion of GIT. 5FU from the two formulations in the stomach and small intestine soaked in a much lesser amount, and a larger dose of the drug was available in the colon region. Elimination half and AUC of FA-CS-5FU-NP-Bo and FA-CS-5FU-NPs was significantly higher (ANOVA, *p* < 0.05) than 5FU solution. The short half-life and AUC of 5FU solution result in absorption of 5FU in the stomach and small intestine.

Consequently, significantly lower (ANOVA, *p* < 0.05) AUC of FA-CS-5FU-NP-Bo than FA-CS-5FU-NPs gives beads formulation of FA-CS-5FU-NP-Bo a comparatively sustained and gradual release of 5FU. These results showed that bare nanoparticles and embedded nanoparticles could delay the release of 5FU in vivo. The results were also consistent with the in vitro release of 5FU.

### 3.6. Biodistribution Studies

In this study, biodistribution of 5FU was carried out in rats model (Figure 9b). The concentration of 5FU solution was greater in liver, heart, and kidney among the three formulations, whereas plasma drug concentration of 5FU solution in colon at 12 h after administration was not detectable. FA-CS-5FU-NPs showed greater distribution in heart, liver, and kidney as compared to FA-CS-5FU-NP-Bo. FA-CS-5FU-NP-Bo showed significantly greater distribution in colon as compared to other organs (ANOVA; *p* < 0.05) (Figure 8b). Kidneys also had greater distribution from 5FU solution and FA-CS-5FU-NPs as compared to the beads formulation. The level of plasma drug concentration was found to be in the following order: 5FU > FA-CS-5FU-NPs > FA-CS-5FU-NP-Bo, which indicated that FA-CS-5FU-NP-Bo reduced more rapid clearance from the circulation than FA-CS-5FU-NPs and 5FU solution due to slower drug-release rate. This result indicated that FA-CS-5FU-NP-Bo improved the colon-distribution characteristics of 5FU as compared to FA-CS-5FU-NPs and 5FU solution.

## 4. Conclusions

Nanoparticles were prepared successfully by the ionic gelation method. FA-CS-5FU-NPs and CS-5FU-NPs were characterized for size and zeta potential and FT-IR, and the study confirmed the conjugation of folic acid with the nanoparticles. Using FA-CS-5FU-NPs alone could not completely avoid the release of 5FU in the stomach and small intestine. Therefore, to prevent the release of 5FU in the upper portion of GIT, FA-CS-5FU-NPs were successfully embedded into calcium alginate beads. Two different drying methods produced FA-CS-5FU-NP-Bf and FA-CS-5FU-NP-Bo. SI, EI, and WUI of FA-CS-5FU-NP-Bf and FA-CS-5FU-NP-Bo depicted significant differences. At acidic pH, there was no release of 5FU from both types of beads. At simulated intestinal pH, percent release rates of 5FU from both FA-CS-5FU-NP-Bf and FA-CS-5FU-NP-Bo beads ranged between 39.3 ± 2 and 96.5 ± 1 and 20.2 ± 2 to 95.9 ± 3, respectively. The high swelling behavior of FA-CS-5FU-NP-Bf beads gives a significantly high release rate. Different pharmacokinetic parameters, such as C_max_, T_max_, T_1/2_, Vd, and AUC_0–t_, for FA-CS-5FU-NP-Bo, FA-CS-5FU-NPs, and 5FU solution were determined. The significantly higher AUC _0-t_ and longer T_1/2_ of FA-CS-5FU-NPs and FA-CS-5FU-NP-Bo compared to the 5FU solution suggest their sustained and site-specific release of 5FU. The accumulation of drug from FA-CS-5FU-NP-Bo was higher in colon as compared to 5FU solution and nanoparticles. Consequently, the use of sodium alginate as an excipient for encapsulation of nanoparticles made it possible to design such a formulation that avoids the premature release of the drug before reaching the colon region. Such formulation can be used for drug delivery to colon-specific sites.

## Figures and Tables

**Figure 1 pharmaceutics-14-01366-f001:**
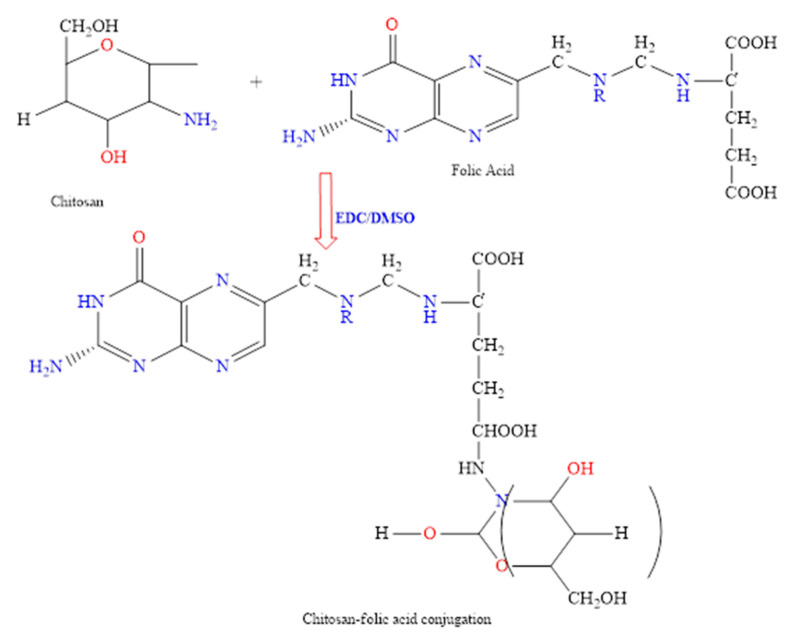
A scheme illustrating the reaction of chitosan with folic acid.

**Figure 2 pharmaceutics-14-01366-f002:**
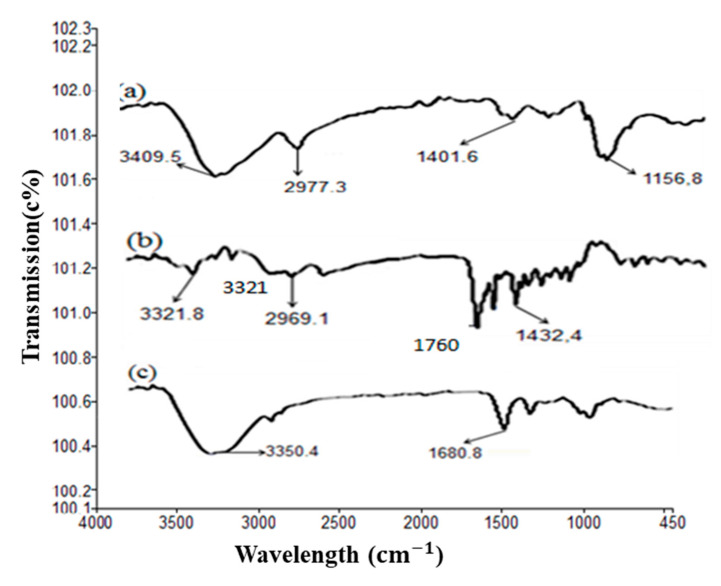
FT-IR illustration of the conjugation of FA with chitosan: (a) pure chitosan, (b) pure FA, and (c) FA-CS conjugate.

**Figure 3 pharmaceutics-14-01366-f003:**
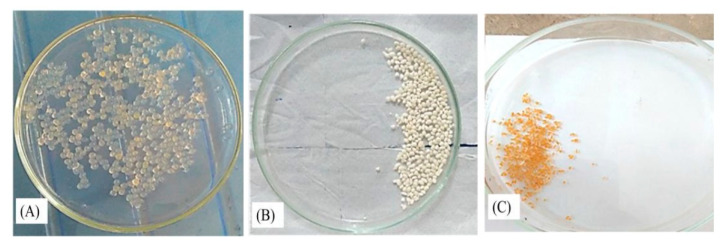
Schematic diagram representing (**A**) freshly prepared beads before drying, (**B**) freeze-dried beads, and (**C**) oven-dried beads.

**Figure 4 pharmaceutics-14-01366-f004:**
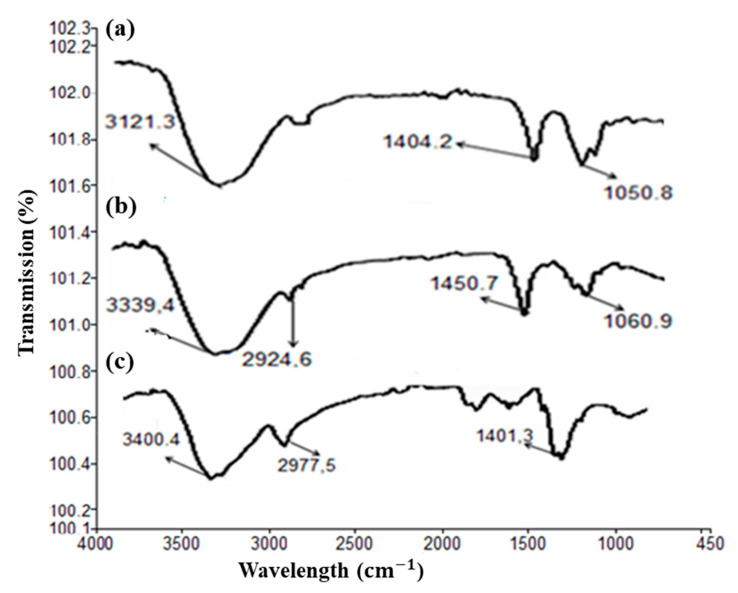
FT-IR of alginate (a), FA-CS-5FU-NPs beads, (b) and pure chitosan (c).

**Figure 5 pharmaceutics-14-01366-f005:**
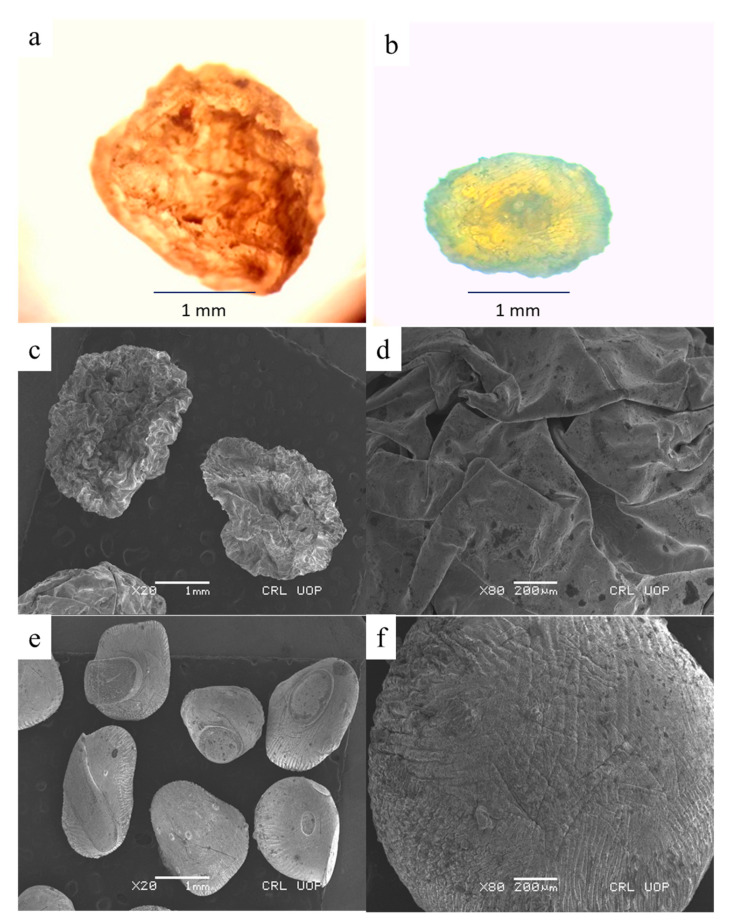
Surface morphology of (**a**) FA-CS-5FU-NP-Bf, (**b**) FA-CS-5FU-NP-Bo, (**c**,**d**) SEM FA-CS-5FU-NP-Bf, and (**e**,**f**) FA-CS-5FU-NP-Bo.

**Figure 6 pharmaceutics-14-01366-f006:**
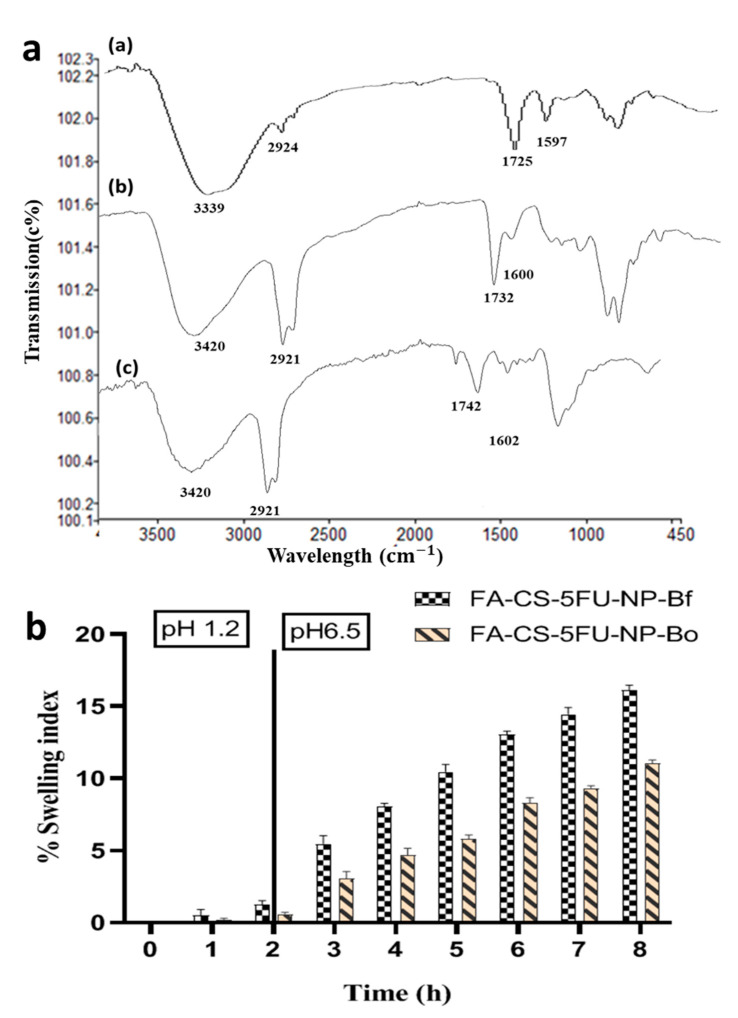
(**a**) FT-IR of swelling study: (a) beads formulation before immersion into fluid medium, (b) beads immersed in a simulated gastric fluid medium, (c) beads immersed into the simulated intestinal fluid medium, and (**b**) swelling behavior of FA-CS-5FU-NP-Bf and FA-CS-5FU-NP-Bo.

**Figure 7 pharmaceutics-14-01366-f007:**
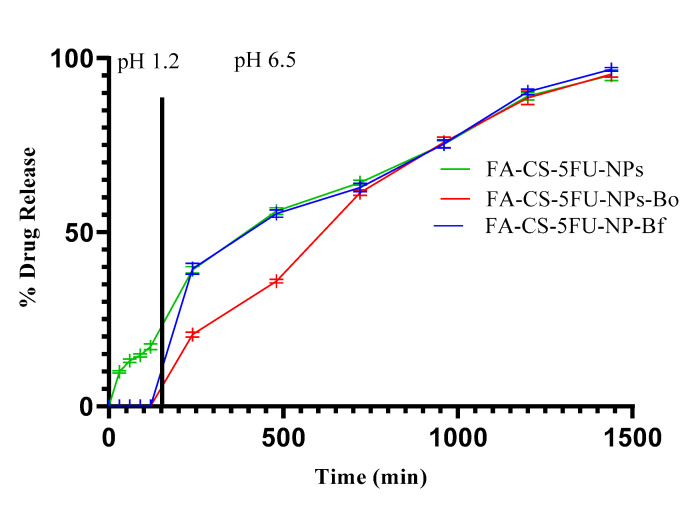
In vitro drug release study of FA-CS-5FU-NPs and FA-CS-5FU-NP-Bf and FA-CS-5FU-NP-Bo.

**Figure 8 pharmaceutics-14-01366-f008:**
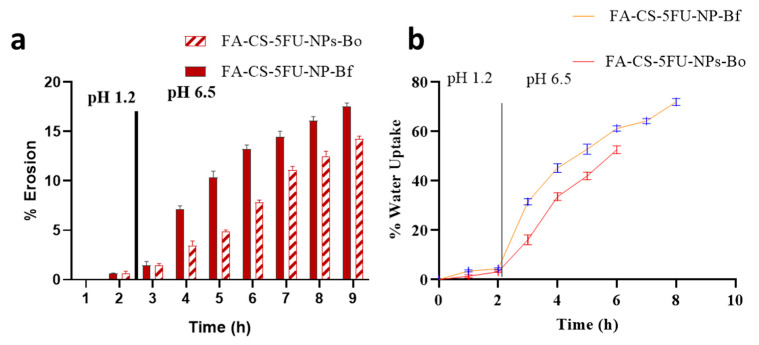
(**a**) Erosion of FA-CS-5FU-NP-Bf and FA-CS-5FU-NP-Bo; (**b**) water-uptake index of FA-CS-5FU-NP-Bf and FA-CS-5FU-NP-Bo.

**Figure 9 pharmaceutics-14-01366-f009:**
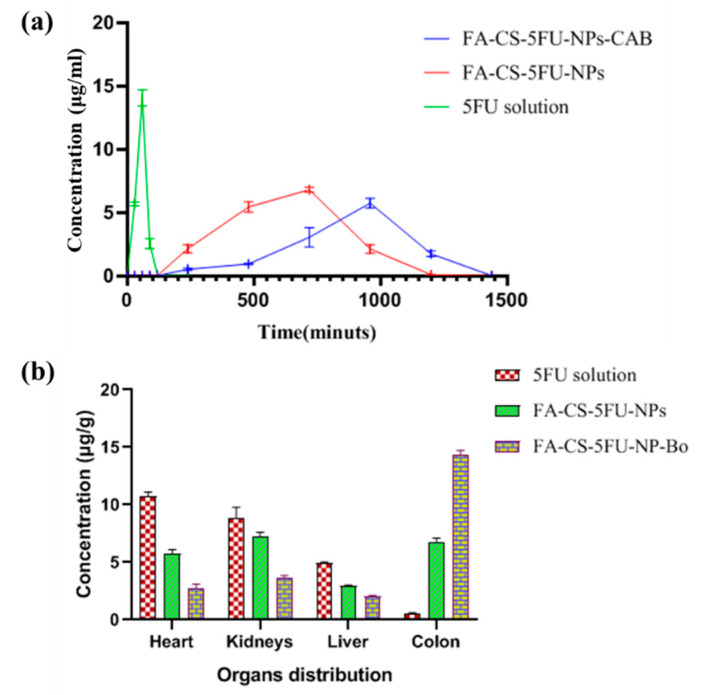
(**a**) Pharmacokinetic study and (**b**) biodistribution study.

**Table 1 pharmaceutics-14-01366-t001:** Mean diameter and shape of FA-CS-5FU-NP-Bf and FA-CS-5FU-NP-Bo.

Beads	Size (mm)	Shape	
		SF	AR
FA-CS-5FU-NP-Bf	2.1 ± 0.07	0.16 ± 0.03	1.4 ± 1.01
FA-CS-5FU-NP-Bo	1.12 ± 0.2	0.17 ± 0.07	1.5 ± 1.2

Data are expressed as mean ± SD; *n* = 3.

**Table 2 pharmaceutics-14-01366-t002:** Percent EE, drug content, and yield of FA-CS-5FU-NP-Bf and FA-CS-5FU-NP-Bo.

Types of Beads	Entrapment Efficiency (%)	Drug Content (%)	Yield (%)
FA-CS-5FU-NP-Bf	96.2 ± 2.21	43 ± 2.03	86.5
FA-CS-5FU-NP-Bo	92 ± 2.43	38.5 ± 2.14	86.5

Data are expressed as mean ± SD; *n* = 3.

**Table 3 pharmaceutics-14-01366-t003:** Pharmacokinetic parameters.

PK Parameters	5FU Solution	FA-CS-5FU-NPs	FA-CS-5FU-NP-B
C_max_ (µg)	14.03 ± 1.52	6.9 h ± 1.21	5.7 ± 0.71
T_max_ (h)	1 h ± 0.17	9 h ± 0.94	12 ± 0.32
T_1/2_ (h)	0.27 ± 0.32	6.3 ± 0.12	7.7 ± 0.20
Ke (1/h)	2.5 ± 0.91	0.11 ± 0.31	0.09 ± 0.05
Vd (L)	0.9 ± 0.05	1.9 ± 0.02	2.1 ±0.01
AUC _0–t_ (µg·h/mL)	18.7 ± 0.65	64.1 ± 1.93	47 ± 1.11

## Data Availability

Not applicable.

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
