# Peer review of "Folate-Modified Chitosan 5-Flourouraci Nanoparticles-Embedded Calcium Alginate Beads for Colon Targeted Delivery"

_pharmaceutics, 2022, doi:10.3390/pharmaceutics14071366_

Round 1
Reviewer 1 Report
SIGNIFICANCE OF THE WORK:
In this paper, embed folate-modified chitosan 5-FU nanoparticles (FA-CS-5FU-NPs) into calcium alginate beads for colon targeted delivery have been prepared. Additionaly, harmacokinetic parameters of 5-FU were analyzed in vivo.
TEXT:
L74, 75: Capital letters:” -Ethyl-3-(3-dimethyl aminopropyl) carbodiimide”, “Dimethylsulfoxide”
Please, improve the quality of all figures
L320: “peak” band
COMMENTS:
This paper is an extension of the one published recently in Polymers (doi.org/10.3390/polym14102010). In this paper, (5-Fluorouracil-Loaded Folic-Acid-Fabricated Chitosan Nanoparticles for Site-Targeted Drug Delivery Cargo,) is described the preparation of the particles. In the present manuscript such nanoparticles are encapsulated into calcium alginate. Although in vivo experiments are performed, only a pharmacokinetic study is presented. In my opinion, the biodistribution of the drug and its effect in some animal model must be described at this stage of the research.
Author Response
Greetings of the day Dear Reviewer!
First of all, I am really indebted to you for your kind consideration of our research article. The valuable suggestions from your kind-self regarding our research article reflected your thorough grounding and expertise in the specialized filed for which I am much grateful to you.
SIGNIFICANCE OF THE WORK:
In this paper, embed folate-modified chitosan 5-FU nanoparticles (FA-CS-5FU-NPs) into calcium alginate beads for colon targeted delivery have been prepared. Additionaly, harmacokinetic parameters of 5-FU were analyzed in vivo.
TEXT:
L74, 75: Capital letters:” -Ethyl-3-(3-dimethyl aminopropyl) carbodiimide”, “Dimethylsulfoxide”
Response: Thanks for comments. Corrections were made accordingly.
Please, improve the quality of all figures
Response: Thanks for suggestions. Figures were improved and scale was added where required.
L320: “peak” band
Response: Thanks for comments. Corrections were made accordingly.
COMMENTS:
This paper is an extension of the one published recently in Polymers (doi.org/10.3390/polym14102010). In this paper, (5-Fluorouracil-Loaded Folic-Acid-Fabricated Chitosan Nanoparticles for Site-Targeted Drug Delivery Cargo,) is described the preparation of the particles. In the present manuscript such nanoparticles are encapsulated into calcium alginate. Although in vivo experiments are performed, only a pharmacokinetic study is presented. In my opinion, the biodistribution of the drug and its effect in some animal model must be described at this stage of the research.
Response: Thanks for your suggestion. Biodistribution study of 5-FU solution, nanoparticles and nanoparticles embedded beads was incorporated in the manuscript.
Reviewer 2 Report
Dear authors,
The presented manuscript is characterized by a scientific soundness. The idea of the study is scientifically significant and suggests achieving of the aimed targeted pH-dependent delivery. The topic will be of interest to the readers. However, there are some concerns about the manuscript and several mistakes that have to be corrected:
Point 1: The English language requires extensive grammatical and lexical corrections. I would suggest to the authors to ask a native English speaker to correct the English or to search for the help of professional English editors.
Point 2: The introduction should be expanded to include more information on the use of folate acide for modification of 5-FU. It also should include information if the authors are the first to prepare folate modified chitosan 5-FU nanoparticles and to include them into alginate beads or there are also other researches who worked on such modifications? The introduction should include a more comprehensive review with summary and analysis of the available research data on the use of such kind of nanoparticles with pH-dependent release and to lead the readers step by step to the hypothesis of the authors thereby pointing to the gaps in the known scientific facts. What is the novelty of this study? It is not clearly highlighted.
Point 3: There is no clearly formulated aim of the study which should be included at the end of the introduction. The aim should underline the originality of the study.
Point 4: The authors need to put more emphasis on the bioadhesivity of the nanoparticles which is surely important for the aim of the study and more information and discussion about this will increase its scientific soundness.
Point 5: There is no scaling on the X-axes in Figures 2, 3 and 5. Usually the original graphs of the instrument should have a scale, printed by the software of the instrument. Why there is no scale here, having in mind that in your previous articles there is a scale on this type of analysis? I would suggest to present the original results (scaled graphs) at least as supplementary material.
Point 6: The authors found out that FA-CS-5FU-NP-Bf beads are disintegrated by 2 hours slower than FA-CS-5FU-NP-Bo beads. On the other side, they have a higher SI and slightly higher percentage of erosion. How could the authors explain this?
Point 7: The pH in the in vitro release study was 2.1 (2.6.6) or 1.2 (3.4.5, Fig. 7)?
Other remarks:
Lines 106, 107 and 123 – dropwise is usually written as one word.
Line 146 – width or breadth?
Line 223 – missing text
Line 347 – beads or beadings?
If the authors undertake the suggested corrections and correct the English, it could be acceptable for publication.
Kind regards,
the Reviewer
Author Response
Greetings of the day Dear Reviewer!
First of all, I am really indebted to you for your kind consideration of our research article. The valuable suggestions from your kind-self regarding our research article reflected your thorough grounding and expertise in the specialized filed for which I am much grateful to you.
Please find below stepwise reply for your suggested comments.:
Dear authors,
The presented manuscript is characterized by a scientific soundness. The idea of the study is scientifically significant and suggests achieving of the aimed targeted pH-dependent delivery. The topic will be of interest to the readers. However, there are some concerns about the manuscript and several mistakes that have to be corrected:
Point 1: The English language requires extensive grammatical and lexical corrections. I would suggest to the authors to ask a native English speaker to correct the English or to search for the help of professional English editors.
Response: Thanks for suggestions. The grammatical errors were corrected. We will seek help of English editors from MDPI English editors.
Point 2: The introduction should be expanded to include more information on the use of folate acide for modification of 5-FU. It also should include information if the authors are the first to prepare folate modified chitosan 5-FU nanoparticles and to include them into alginate beads or there are also other researches who worked on such modifications? The introduction should include a more comprehensive review with summary and analysis of the available research data on the use of such kind of nanoparticles with pH-dependent release and to lead the readers step by step to the hypothesis of the authors thereby pointing to the gaps in the known scientific facts. What is the novelty of this study? It is not clearly highlighted.
Response: Thanks for your comments and suggestions. Information about folic acid summary and review of available research data about NPs with pH sensitive release were included in the introduction section.
- Work has been done on folate modified chitosan Nanoparticles but coating of such Nanoparticles with alginate polymer is a novel work (in-order to prevent release of drug in stomach and to get maximum accumulation in colon)
- Coating unconjugated chitosan Nanoparticles with pH sensitive polymers has been investigated in other studies, but coating folic acid conjugated chitosan nanoparticles with alginate polymer is a novel work.
Point 3: There is no clearly formulated aim of the study which should be included at the end of the introduction. The aim should underline the originality of the study.
Response: Thanks for suggestion. Aim of study was incorporated in introduction.
Point 4: The authors need to put more emphasis on the bioadhesivity of the nanoparticles which is surely important for the aim of the study and more information and discussion about this will increase its scientific soundness.
Response: Thanks for your suggestion. Dear sir, this work mainly on synthesis of FA-CS conjugate, nanoparticles and NP embedded beads for colon delivery. Your suggestion is valuable and we will consider in our further studies.
Point 5: There is no scaling on the X-axes in Figures 2, 3 and 5. Usually the original graphs of the instrument should have a scale, printed by the software of the instrument. Why there is no scale here, having in mind that in your previous articles there is a scale on this type of analysis? I would suggest to present the original results (scaled graphs) at least as supplementary material.
Response: Thanks for suggestions. Scale was included in above mentioned figures.
Point 6: The authors found out that FA-CS-5FU-NP-Bf beads are disintegrated by 2 hours slower than FA-CS-5FU-NP-Bo beads. On the other side, they have a higher SI and slightly higher percentage of erosion. How could the authors explain this?
Response: Thanks for comments. The reason for slower disintegration of FA-CS-5FU-NP-Bf is its stable internal ionic gelation due to its porous nature than external ionic gelation of Non porous FA-CS-5FU-NP-Bo Beads.
Point 7: The pH in the in vitro release study was 2.1 (2.6.6) or 1.2 (3.4.5, Fig. 7)?
Response: Thanks for comments. The pH was corrected to 1.2.
Other remarks:
Lines 106, 107 and 123 – dropwise is usually written as one word.
Line 146 – width or breadth?
Line 223 – missing text
Line 347 – beads or beadings?
Response: Thanks for suggestions. All the corrections were made accordingly.
If the authors undertake the suggested corrections and correct the English, it could be acceptable for publication.
Kind regards,
the Reviewer
Reviewer 3 Report
I have gone through the manuscript "Folate Modified Chitosan 5FU Nanoparticles Embedded Calcium Alginate Beads for Colon Targeted Delivery". I have some suggestions to improve the quality and readability of the manuscript.
1. The novelty of this study needs to define clearly in the introduction.
2. It will be great if the authors could explain line 78, how chitosan conjugates with FA at low pH, one representative pictorial figure will be a nice addition.
3. It will be nice if the authors could mention the molecular weight of Chitosan and alginate used in this study.
4. To have a clear conjugation process, please mention the degree of the substitution.
5. After conjugation the CS-FA solubility should change, it does not need low pH to solubilize, please explain why you have chosen low pH for the solubility.
6. Some of the figure quality need to be improved.
7. Try to add some recent references, if possible include some references from Pharmaceutics journal.
Author Response
Dear Reviewer!
I acknowledge your efforts regarding sparing your precious time to evaluate our research article and gave valuable assessment to make it publishable. Yours comments are commendable and helped us a lot to refine this manuscript as well as will help us in future to keep in mind all these parameters to make the research articles acceptable in each and every aspect. After proper thanking you it’s to bring into your kind attention that the entire manuscript has been thoroughly checked and all the spelling errors and other corrections have been corrected carefully.
I have gone through the manuscript "Folate Modified Chitosan 5FU
Nanoparticles Embedded Calcium Alginate Beads for Colon Targeted
Delivery". I have some suggestions to improve the quality and
readability of the manuscript.
1. The novelty of this study needs to define clearly in the introduction.
Response. Thanks for suggestions. The novelty of the study was clearly defined in introduction.
2. It will be great if the authors could explain line 78, how chitosan
conjugates with FA at low pH, one representative pictorial figure will
be a nice addition.
Response. Thanks for suggestions. Chitosan is polycationic in acidic media (pKa 6.5) and can interact with negatively charged species such as folic acid. Added figure in the manuscript to explain conjugation.
3. It will be nice if the authors could mention the molecular weight of
Chitosan and alginate used in this study.
Response. Thanks for suggestions. Chitosan (DD, 83% and mol wt 310000) and sodium alginate (Mw=710,974). Mentioned all these informations in materials section of the manuscript.
4. To have a clear conjugation process, please mention the degree of the
substitution.
Response. Thanks for suggestions. Degree of substitution was mentioned it in the manuscript
5. After conjugation the CS-FA solubility should change, it does not
need low pH to solubilize, please explain why you have chosen low pH for
the solubility.
Response. Thanks for comments. The pH of the conjugation still remain acidic, so, conjugates soluble at low pH.
6. Some of the figure quality need to be improved.
Response. Thanks for comments. Figures quality was improved.
7. Try to add some recent references, if possible include some
references from Pharmaceutics journal.
Response. Thanks for suggestions. New references including reference from pharmaceutics journal was also added in the final version of manuscript.
Round 2
Reviewer 1 Report
Although I still think the presentation must be improved and the efficiency of the delivery system should be tested in mice models, the authors have response essentially to my recommendations.
Author Response
Dear Reviewer!
I acknowledge your efforts regarding sparing your precious time to evaluate our research article and gave valuable assessment to make it publishable. Yours comments are commendable and helped us a lot to refine this manuscript as well as will help us in future to keep in mind all these parameters to make the research articles acceptable in each and every aspect. After proper thanking you it’s to bring into your kind attention that the entire manuscript has been thoroughly checked and all the spelling errors and other corrections have been corrected carefully.
Although I still think the presentation must be improved and the efficiency of the delivery system should be tested in mice models, the authors have response essentially to my recommendations.
Response: Thanks for your suggestions. This study focused on to minimize release of drug in stomach and maximum drug is available in colon region. These findings were confirmed via in vitro and in vivo experiments. The results were encouraging. Induction of colon cancer in rats was not performed at this stage. In future this system will be carried out.
Reviewer 2 Report
Dear authors,
Almost all my remarks were taken into consideration and the manuscript is now significantly improved. However, there are still some minor corrections to be done:
Point 1: The English language requires extensive grammatical and lexical corrections. I would suggest to the authors to ask a native English speaker to correct the English or to search for the help of professional English editors.
Response: Thanks for suggestions. The grammatical errors were corrected. We will seek help of English editors from MDPI English editors.
Point 1.1: There are still too many grammar mistakes in the text. They have to be corrected so that the manuscript will be acceptable for publication.
Point 2: The introduction should be expanded to include more information on the use of folate acide for modification of 5-FU. It also should include information if the authors are the first to prepare folate modified chitosan 5-FU nanoparticles and to include them into alginate beads or there are also other researches who worked on such modifications? The introduction should include a more comprehensive review with summary and analysis of the available research data on the use of such kind of nanoparticles with pH-dependent release and to lead the readers step by step to the hypothesis of the authors thereby pointing to the gaps in the known scientific facts. What is the novelty of this study? It is not clearly highlighted.
Response: Thanks for your comments and suggestions. Information about folic acid summary and review of available research data about NPs with pH sensitive release were included in the introduction section.
- Work has been done on folate modified chitosan Nanoparticles but coating of such Nanoparticles with alginate polymer is a novel work (in-order to prevent release of drug in stomach and to get maximum accumulation in colon)
- Coating unconjugated chitosan Nanoparticles with pH sensitive polymers has been investigated in other studies, but coating folic acid conjugated chitosan nanoparticles with alginate polymer is a novel work.
Point 2.1: Thank you to the authors for considering my suggestions. The introduction is really improved by the addition of more comprehensive information about the use of folate acide and pH-depend systems for drug release. There are still some minor improvments to be done: Paragraph two of the introduction (lines 73-75) is not a smooth continuation of the previous one and the information is not arranged in a logical sequence. The first sentence is more related to the purpose, while the second would be a direct continuation of the previous paragraph, but in this case the thought is interrupted by the previous sentence. The sentences in this paragraph should be logically distributed throughout the text.
Point 4: The authors need to put more emphasis on the bioadhesivity of the nanoparticles which is surely important for the aim of the study and more information and discussion about this will increase its scientific soundness.
Response: Thanks for your suggestion. Dear sir, this work mainly on synthesis of FA-CS conjugate, nanoparticles and NP embedded beads for colon delivery. Your suggestion is valuable and we will consider in our further studies.
Point 4.1: I understand the point of the authors and there is no need of further experiments here. My suggestion was to discuss the bioadhesivity of the carrier as far as it can improve the targeted drug delivery and is actually an advantage of your system. This information should be mentioned in the text – in the introduction for example.
Point 5: There is no scaling on the X-axes in Figures 2, 3 and 5. Usually the original graphs of the instrument should have a scale, printed by the software of the instrument. Why there is no scale here, having in mind that in your previous articles there is a scale on this type of analysis? I would suggest to present the original results (scaled graphs) at least as supplementary material.
Response: Thanks for suggestions. Scale was included in above mentioned figures.
Point 5.1: Figure 6 is still without a scale.
Other remarks:
Line 390 - The authors use “beads” throughout the whole text and only here they use “beadings”? Shouldn’t they use beads also here?
Kind regards,
the Reviewer
Author Response
Dear authors,
Almost all my remarks were taken into consideration and the manuscript is now significantly improved. However, there are still some minor corrections to be done:
Point 1: The English language requires extensive grammatical and lexical corrections. I would suggest to the authors to ask a native English speaker to correct the English or to search for the help of professional English editors.
Response: Thanks for suggestions. The grammatical errors were corrected. We will seek help of English editors from MDPI English editors.
Point 1.1: There are still too many grammar mistakes in the text. They have to be corrected so that the manuscript will be acceptable for publication.
Point 2: The introduction should be expanded to include more information on the use of folate acide for modification of 5-FU. It also should include information if the authors are the first to prepare folate modified chitosan 5-FU nanoparticles and to include them into alginate beads or there are also other researches who worked on such modifications? The introduction should include a more comprehensive review with summary and analysis of the available research data on the use of such kind of nanoparticles with pH-dependent release and to lead the readers step by step to the hypothesis of the authors thereby pointing to the gaps in the known scientific facts. What is the novelty of this study? It is not clearly highlighted.
Response: Thanks for your comments and suggestions. Information about folic acid summary and review of available research data about NPs with pH sensitive release were included in the introduction section.
- Work has been done on folate modified chitosan Nanoparticles but coating of such Nanoparticles with alginate polymer is a novel work (in-order to prevent release of drug in stomach and to get maximum accumulation in colon)
- Coating unconjugated chitosan Nanoparticles with pH sensitive polymers has been investigated in other studies, but coating folic acid conjugated chitosan nanoparticles with alginate polymer is a novel work.
Point 2.1: Thank you to the authors for considering my suggestions. The introduction is really improved by the addition of more comprehensive information about the use of folate acide and pH-depend systems for drug release. There are still some minor improvments to be done: Paragraph two of the introduction (lines 73-75) is not a smooth continuation of the previous one and the information is not arranged in a logical sequence. The first sentence is more related to the purpose, while the second would be a direct continuation of the previous paragraph, but in this case the thought is interrupted by the previous sentence. The sentences in this paragraph should be logically distributed throughout the text.
Point 4: The authors need to put more emphasis on the bioadhesivity of the nanoparticles which is surely important for the aim of the study and more information and discussion about this will increase its scientific soundness.
Response: Thanks for your suggestion. Dear sir, this work mainly on synthesis of FA-CS conjugate, nanoparticles and NP embedded beads for colon delivery. Your suggestion is valuable and we will consider in our further studies.
Point 4.1: I understand the point of the authors and there is no need of further experiments here. My suggestion was to discuss the bioadhesivity of the carrier as far as it can improve the targeted drug delivery and is actually an advantage of your system. This information should be mentioned in the text – in the introduction for example.
Point 5: There is no scaling on the X-axes in Figures 2, 3 and 5. Usually the original graphs of the instrument should have a scale, printed by the software of the instrument. Why there is no scale here, having in mind that in your previous articles there is a scale on this type of analysis? I would suggest to present the original results (scaled graphs) at least as supplementary material.
Response: Thanks for suggestions. Scale was included in above mentioned figures.
Point 5.1: Figure 6 is still without a scale.
Other remarks:
Line 390 - The authors use “beads” throughout the whole text and only here they use “beadings”? Shouldn’t they use beads also here?
Kind regards,
the Reviewer